# The Novel Monoclonal IgG_1_-Antibody AB90-E8 as a Diagnostic Tool to Rapidly Distinguish *Aspergillus fumigatus* from Other Human Pathogenic *Aspergillus* Species

**DOI:** 10.3390/jof9060622

**Published:** 2023-05-28

**Authors:** Tamara Katharina Kakoschke, Christoph Kleinemeier, Thomas Knösel, Sara Carina Kakoschke, Frank Ebel

**Affiliations:** 1Department of Oral and Maxillofacial Surgery and Facial Plastic Surgery, University Hospital, Ludwig-Maximilians-University Munich, Lindwurmstrasse 2a, 80337 Munich, Germany; 2Institute for Infectious Diseases and Zoonoses, Ludwig-Maximilians-University Munich, 85764 Oberschleissheim, Germanyfrank.ebel@lmu.de (F.E.); 3Institute of Pathology, Ludwig-Maximilians-University Munich, 81377 Munich, Germany; 4Department of General, Visceral and Transplant Surgery, University Hospital, Ludwig-Maximilians-University Munich, Marchioninistrasse 15, 81337 Munich, Germany

**Keywords:** invasive aspergillosis, *Aspergillus fumigatus*, monoclonal antibody, fungal diagnostics

## Abstract

In most cases, invasive aspergillosis (IA) is caused by *A. fumigatus*, though infections with other *Aspergillus* spp. with lower susceptibilities to amphotericin B (AmB) gain ground. *A. terreus*, for instance, is the second leading cause of IA in humans and of serious concern because of its high propensity to disseminate and its in vitro and in vivo resistance to AmB. An early differentiation between *A. fumigatus* and *non-A. fumigatus* infections could swiftly recognize a potentially ineffective treatment with AmB and lead to the lifesaving change to a more appropriate drug regime in high-risk patients. In this study, we present the characteristics of the monoclonal IgG_1_-antibody AB90-E8 that specifically recognizes a surface antigen of *A. fumigatus* and the closely related, but not human pathogenic *A. fischeri*. We show immunostainings on fresh frozen sections as well as on incipient mycelium picked from agar plates with tweezers or by using the expeditious tape mount technique. All three methods have a time advantage over the common procedures currently used in the routine diagnosis of IA and outline the potential of AB90-E8 as a rapid diagnostic tool.

## 1. Introduction

Invasive aspergillosis (IA) is the most common form of invasive pulmonary mold disease among immunocompromised patients and is associated with a high morbidity and mortality rate [1]. Predominant risk factors for invasive fungal diseases are haematological malignancies, prolonged (>10 days) neutropenia (especially when associated with fever of unknown origin ≥ 72 h), allogenic stem cell transplantation, solid organ transplantation, medication-associated immunosuppression with corticosteroids, T-cell or B-cell immunosuppressants, acute graft-versus-host disease and inherited severe immunodeficiencies [2]. Yet IA has also a rising relevance in non-neutropenic patients, i.e., with chronic obstructive pulmonary disease (COPD), and in ICU patients suffering from viral infections, such as influenza and COVID-19 disease [3,4,5,6].

The dominant causative subspecies of IA is *A. fumigatus*, but other human pathogenic *Aspergillus* spp. also occur and are of clinical relevance.

Thus, *A. terreus*, *A. niger* and *A. flavus* were isolated from transplant patients [7,8], and *A. nidulans* was found to be the second leading pathogen causing invasive fungal disease in chronic granulomatous disease, a hereditary immunodeficiency [9]. Worldwide, infections with *A. terreus* were found to be rising with over 5% of mold infections collected during a one-year study in 2014/2015 covering 21 countries [10]. 39.2% of these *A. terreus* isolates were found in patients suffering from chronic lung diseases [10].

In contrast to *A. fumigatus*, *A. terreus* strains generally have an intrinsic resistance to the antifungal drug amphotericin B (AmB) [11]. In Tyrol (Austria), a hotspot of *A. terreus* infections [12,13], 98% of the clinical *A. terreus* isolates were resistant to AmB [14]. As liposomal AmB still belongs to the first-line treatment of aspergillosis (alternative to azoles) [15], the increasing impact of *A. terreus* infections might be promoted by frequent use of AmB. In addition, other non-fumigatus *Aspergillus* species, such as *A. nidulans* and *A. flavus*, showed higher MICs to AmB than *A. fumigatus* [16].

Among the human pathogenic *Aspergillus* species there are differences not only in their resistance levels but also in their pathogenesis. This poses special risks concerning dissemination in affected patients. Compared to *A. fumigatus*, *A. terreus* needs a greater fungal load to cause infection [17,18], and in vitro experiments revealed a faster phagocytosis of *A. terreus* conidia compared to *A. fumigatus* conidia in macrophages [19]. However, while *A. fumigatus* conidia were mostly inactivated (40–70% within 10 h) after phagocytosis, and only the surviving residuals germinate fast and penetrated the immune cells from within, the *A. terreus* conidia persisted and remained dormant in macrophages [19] and survived longer intracellularly [20]. This and the special properties of accessory conidia, which are unique to *A. terreus* [21,22], may contribute to the high rate of dissemination in *A. terreus* infections. It also correlates with stronger antifungal drug resistance and points out the specific challenges in treating these infections. The findings also suggest that the uptake of *A. terreus* spores might take place long before hospitalization [12,20].

Not least in patients with a contraindication for azoles (i.e., interactions with anticancer drugs [23]), an early differentiation between *A. fumigatus* with higher AmB susceptibility and other *Aspergillus* spp. with low AmB susceptibility can accelerate and improve therapy by early drug changes and save lives in high-risk patients and fulminant infections.

In this study, we present simple diagnostic techniques that allow an easy and expeditious differentiation of *A. fumigatus* from other human pathogenic *Aspergillus* spp. using the novel antibody AB90-E8. This is a monoclonal IgG_1_ antibody that was recently used to generate chimeric antigen receptor (CAR)-expressing T cells to treat invasive pulmonary aspergillosis in preclinical mouse models of infection [24]. In this article, we provide additional information about further characteristics of the monoclonal antibody AB90-E8 and present its diagnostic benefits in achieving a faster and unambiguous identification of *A. fumigatus* in clinical samples.

## 2. Materials and Methods

### 2.1. Strains and Media

The following strains were used in this study: *A. fumigatus* ATCC46645 [25], *A. fumigatus* CEA17 [26], *A. fumigatus* Af293 [27,28], *A. fumigatus* D141 [29], *A. fumigatus* AfS35 [30] and GFP-expressing AfS35 [31], *A. fumigatus* D141∆*glf*A (unable to produce galactofuranose) and the complemented strain D141∆*glf*A+*glf*A [32], *A. fischeri* DSM 3700 [33], *A. terreus* T9 [34], *A. terreus* SBUG 844 [17,19], *A. terreus* FGSC A1156 (=NIH2624), *A. niger* DSM 737 [35,36], *A. niger* CBS 513.88 [37], *A. nidulans* DSM 820 (=ATCC11267), *A. nidulans* FGSC A26, *A. oryzae* DSM 1863 and *A. giganteus* DSM 1146. Additionally, nine different clinical *A. fumigatus* isolates from human origin, five different clinical *A. fumigatus* isolates from horses, ten different clinical *A. fumigatus* isolates from dogs and one clinical *A. terreus* isolate with canine origin were used in this study. The clinical human isolates were obtained from the Max-von-Pettenkofer Institute (LMU Munich, Munich, Germany) and the clinical animal isolates from the Institute for Infectious Diseases and Zoonoses (LMU Munich). Each identification was confirmed by PCR and subsequent sequencing. Strains were routinely grown on Sabouraud agar or in Sabouraud liquid medium at 37 °C or 30 °C. When indicated, the following other media were used: Aspergillus minimal medium (AMM) (0.52 g KCl, 0.52 g MgSO_4_ × 7H_2_O, 1.52 g KH_2_PO_4_, 1 mL essential element solution (40 mg Na_2_B_4_O_7_ × 10H_2_O, 400 mg CuSO_4_ × 3H_2_O, 800 mg FePO_4_ × H_2_O, 800 mg MnSO_4_ × 4H_2_O, 800 mg Na_2_MoO_4_ × 2H_2_O, 8 g ZnSO_4_ × 7H_2_O; ad 1 L; autoclaved), 10 g glucose, sodium nitrate 6 g, pH 6.5; ad 1 L), yeast glucose (YG) medium (5 g/L yeast extract, 20 g/L glucose), RPMI-1640 (Gibco, Thermo Fisher Scientific) buffered with 20 mM HEPES, BSA medium (PBS supplemented with 2 g/L bovine serum albumin) and synthetic defined (SD) medium (1.7 g/L yeast nitrogen base with 5 g/L ammonium sulfate and 2 g/L D-glucose).

### 2.2. Antibodies for Immunostaining

Hybridoma supernatants of the monoclonal antibodies (mAb) AB90-E8 and L10-1 were used for immunostaining experiments. AB90-E8 belongs to the IgG_1_-class and is specific for *A. fumigatus* and the closely related, but not human-pathogenic *A. fischeri* [24]. The mAB L10-1, which is specific for galactomannan [38], belongs to the IgM-class and was used as a control and in double staining experiments. To gain antibody-saturated hybridoma supernatants, hybridoma cells were grown in OPTI-MEM supplemented with 4% fetal calf serum. The supernatants were harvested by centrifugation (10 min, 500× *g*) when the cells reached the stationary phase.

In this study, the following secondary antibodies were used: the Cy3-labeled anti-mouse IgG+M (visualized in red), the Cy3-labelled anti-mouse IgM (visualized in red) or the Alexa Fluor 488-labeled anti-mouse IgG (H + L) antibody (visualized in green) (Dianova, Hamburg, Germany).

To stain the fungal cell wall, samples were incubated with 500 µg/mL Calcofluor white (CFW) (Sigma-Aldrich, Deisenhofen, Germany) for 2 min at RT (visualized in blue).

The MP1 antiserum was raised in rabbits against purified, His_6_-tagged MP1 protein. The corresponding pre-serum was used as a control.

### 2.3. Staining of Hyphae Grown on Glass Cover Slips

#### 2.3.1. Growth in Liquid Culture and Immunostaining

Hyphae were grown on glass cover slips in the liquid medium specified in the text. The samples were subsequently fixed with 3.7% formaldehyde for 5 min. After washing with PBS, samples were incubated with the first antibody (hybridoma supernatant) for 30 min at 37 °C in a moistened chamber and then washed three times with PBS-T (PBS containing 0.1% Tween 20). After another incubation with the secondary antibody (diluted 1:200 in PBS, final concentration of 2 µg/mL), the samples were washed again three times with PBS-T. When indicated, samples were additionally incubated with 500 µg/mL CFW for 2 min to stain the fungal cell wall and finally washed with PBS-T.

#### 2.3.2. Fludioxonil Treatment

After growth on glass cover slips in liquid AMM medium, the medium was discarded and replaced by liquid AMM medium containing 2 µg/mL fludioxonil (Sigma Aldrich). The samples were further incubated at 37 °C for 4 h, fixed and stained as described in Section 2.3.1.

#### 2.3.3. Growth and Infection of A549 Epithelial Cells

A549 human lung epithelial cells (ATCC CCL-185) were seeded on glass cover slips in RPMI-1640 supplemented with 5% fetal calf serum and incubated in a tissue culture incubator at 37 °C + 5% CO_2_ for 4 days. The medium was then replaced by fresh medium containing *A. fumigatus* conidia and incubated for 8 h at 37 °C + 5% CO_2_. Fixation and immunostaining were then performed as described in Section 2.3.1.

### 2.4. Direct Staining of Hyphae from Agar Plates

#### 2.4.1. Picking Method

Fungal material of small colonies was picked from an agar plate using tweezers. The first colonies appeared after 12 h incubation at 37 °C. The material was then transferred to a 1 mL Eppendorf tube, optionally fixed with 3.7% formaldehyde for 5 min at room temperature (RT) and washed in PBS-T, or it was directly fully covered with AB90-E8 supernatant and incubated for 30 min at RT in an Eppendorf Thermomixer comfort shaking device at 180 rpm or a rotating wheel mixer at 6 rpm (CAT M. Zipperer Rotating/Swaying Mixer). The tube was then centrifugated at 13,000× *g* rpm for 1 min (Eppendorf 5415D centrifuge). The pellet was resuspended and washed in PBS-T using a Vortex Genie 2 and centrifugated again at 13,000× *g* rpm for 1 min. This washing step was performed three times. The fungal material was then taken up in a 1:200 dilution of the secondary antibody and incubated for another 30 min at RT with shaking. Afterwards, it was washed three times with PBS-T as described above. After the last centrifugation step, the supernatant was discarded and the material was transferred to an object slide, mounted with Vectashield Mounting Medium (Vector Laboratories, Burlingame, CA, USA) and covered with a glass cover slide. Hyphal material obtained after growth in liquid cultures with shaking (fungus balls) was transferred to an Eppendorf tube. The immunostaining was performed as described above.

#### 2.4.2. Tape Mount Immunostaining Method

In this study, we used a modified version of the immunostaining method using adhesive tape that was published in [39]. First, a double-sided sticky glue strip from a glue roller (Tesa^®^Roller permanent) was applied on the upper and lower margins of an object slide (Figure 1A). Then a slice of a transparent adhesive tape (Scotch^®^ Crystal Clear Tape 600 or Tesa^®^Film transparent) was held with tweezers and pressed with its sticky side on early fungal colonies of an agar plate (Figure 1B). The tape was then attached with its non-sticky side to the prepared glue roller stripes on the object slide (Figure 1C). While pressing the tape stripe on the agar plate, it was recommended to touch the fungal material with the central part of the tape stripe. The bound fungal material on the adhesive tape strip could easily be identified macroscopically, and the region was marked with a dot. Multiple samples can therefore be analysed on one object slide (Figure 1D). Fifty µL of 10% goat serum or 1% BSA (Fraction V, Biomol, Hamburg, Germany) were applied to the marked areas and incubated for 20 min at RT in a humid chamber. To wash the samples, the blocking solution was replaced with 50 µL of PBS-T that was slowly pipetted up and down for three times and then removed. Fifty µL of the AB90-E8-containing supernatant was added and the sample was incubated at RT for 30 min in a humid chamber. The washing step with PBS-T was repeated three times, before the secondary antibody was applied to the marked areas and incubated at RT for 30 min in a humid chamber. The washing step with PBS-T was performed another three times. The sample was finally stained with 500 µg/mL CFW for 2 min, washed another three times, embedded in mounting medium and covered with a glass cover slip.

### 2.5. Staining of Histological Samples

Anonymized histological slide preparations (cryo-sections or slides of paraffin-embedded tissue) of human aspergillomas were obtained from the Institute of Pathology (LMU Munich). For these samples, a diagnostic PCR was available that proved an infection with *A. fumigatus*. The study of immunohistochemical stainings with monoclonal antibodies against fungal antigens on human histological specimens/preparations was previously reported to the LMU Ethics Committee but evaluated to require no ethical approval (Ref. No. 19-196 KB).

Prior to staining, paraffin slides were washed twice with 100% xylene, twice with 100% ethanol, twice with 95% ethanol, once with 80% ethanol, once with 70% ethanol, once with 50% ethanol and finally twice with H_2_O. Each washing step took 5 min. Cryo-sections were not deparaffinizated and were directly stained. Slides were covered with PBS containing 10% goat serum for 20 min to block unspecific binding sites and then washed in PBS-T three times. After 30 min incubations with the first (hybridoma supernatant of AB90-E8 or L10-1) and secondary antibody, samples were incubated with PBS containing 500 µg/mL CFW for 2 min. After each step, samples were washed three times with PBS-T. All incubation steps were performed in a closed humid chamber at RT. Samples were finally mounted using Vectashield Mounting Medium and covered with glass cover slips.

Grocott silver staining and HE staining were performed according to standard histological techniques for pathological analysis.

### 2.6. Microscopy

After mounting and covering with glass cover slips, the object slides were optionally sealed with nail polish to protect the samples from drying and to prolong their durability, or they were immediately inspected under the microscope without sealing.

Images were captured with a Leica DM 5000 B microscope equipped with a DFC420 C camera and LAS-X-software (Leica Microsystems, Wetzlar, Germany) or with a confocal laser scanning microscope, either a SP5 (Leica Microsystems, Wetzlar, Germany) or a LSM 880 (Zeiss, Jena, Germany).

## 3. Results

### 3.1. Characterization of the Monoclonal Antibody AB90-E8

The hybridoma clone AB90-E8 was raised against hyphal fragments of *A. fumigatus* and strongly labeled the hyphal surface of *A. fumigatus* strain AfS35 [24]. Similar results were obtained with different *A. fumigatus* strains, including reference strains as well as clinical human and animal isolates [24]. These results demonstrate that the AB90-E8 antigen is expressed by many if not all *A. fumigatus* strains. AB90-E8 strongly decorates the hyphal tubes, even at a very early stage of germination, whereas most conidial bodies, resting and swollen conidia as well as conidiophores and phialides were not or hardly stained [24].

The hyphal tips always showed a strong signal, while the rest of the hyphae, albeit recognized by the antibody over its whole surface, showed only slight differences in the staining pattern and intensity depending on the liquid medium used for growth (Figure 2), indicating that the antigen is expressed under different growth conditions. Strikingly, when grown on A549 epithelial cells, there were completely undecorated parts of the hyphal surface often next to the strongly stained tip (Figure 3).

We have previously shown that the mAb AB90-E8 shows no reactivity with other fungi, such as *Candida albicans*, *Fusarium oxysporum*, *Lichtheimia corymbifera*, *Mucor circinelloides*, *Rhizopus arrhizus* and *Scedosporium boydii* [24]. Within the genus *Aspergillus*, the antibody stained, apart from *A. fumigatus,* only the closely related *A. fischeri*, whereas no reactivity in immunostaining was observed with *A. terreus*, *A. nidulans*, *A. niger*, *A. oryzae* and *A. giganteus* [24]. As *A. terreus* is of particular clinical relevance, we tested three different *A. terreus* strains and one clinical isolate that all turned out to be AB90-E8-negative [24]. Therefore AB90-E8 is not suitable for the detection of all pathogenic Aspergilli, but to identify *A. fumigatus* out of them.

The overall staining patterns of *A. fumigatus* hyphae with AB90-E8 strikingly resembled that of the galactomannan-specific antibody L10-1 [38]. However, the fact that non-fumigatus Aspergilli, which possess galactomannan, are not recognized by AB90-E8, excludes the possibility that AB90-E8 recognizes galactomannan. To investigate the relationship of both antigens, we performed a double immunofluorescence labelling. Figure 4 shows AB90-E8 in green and the galactomannan-specific mAB L10-1 in red. Conidial bodies were only weakly labelled, whereas both antigens were prominent on the hyphal cell wall. At higher magnification, an overlay of both channels demonstrates that the spatial distributions of the two antigens are similar, but not identical (Figure 4D).

The partial co-localization of both antigens prompted us to stain a ∆*glf*A mutant that is unable to produce galactofuranose, an essential component of fungal galactomannan [32]. As shown in Figure 5, hyphae of the ∆*glf*A mutant showed a distribution of the AB90-E8 antigen that was clearly distinct from that observed for the complemented mutant and the corresponding wild type strain D141. As expected, the galactomannan-positive hyphae were more or less homogenously labelled by AB90-E8 (Figure 5A’,C’), whereas the labeling of the Δ*glf*A mutant hyphae was weaker and largely focused at the hyphal tips (Figure 5B’). This demonstrates that the anchorage of the AB90-E8 antigen is impaired in the ∆*glf*A mutant.

To explore this further, we stained hyphae after fludioxonil treatment, which triggers a shedding of the galactomannan antigen [40]. Labelling of hyphae with AB90-E8 was clearly reduced after fludioxonil treatment (Figure 6C), but the staining pattern remained more homogenously than that of galactomannan (Figure 6B).

Most fungal surface antigens are either carbohydrates or proteins. AB90-E8 belongs to the IgG_1_ subclass, which is more common for protein antigens. Cell wall carbohydrates, such as galactomannan, are often released by *A. fumigatus* during growth. Immunoblot analysis of culture supernatants revealed smear-like staining patterns with anticarbohydrate antibodies [38]. Such a pattern was not observed with AB90-E8. Digestion of hyphal surface proteins using proteinase K destroyed the AB90-E8 antigen, a finding that strongly suggests that it is a protein [24].

So far, only very few proteins are known to be accessible on the hyphal surface. One example is the MP1 protein. We used a polyclonal antibody raised against recombinant MP1 to test a potential co-localization of both antigens. As shown in Figure 7, AB90-E8 and anti-MP1 showed clearly different patterns. A corresponding pre-immune serum showed no reactivity.

### 3.2. Application of AB90-E8 as a Diagnostic Tool

For direct identification of *A. fumigatus* in clinical samples, material from colonies grown on agar plates or histological samples were used.

#### 3.2.1. Direct Staining of Hyphae from Agar Plates—Picking Method

First, we tested whether AB90-E8 can be used to stain fungal material that was directly picked from colonies on agar plates as described in Section 2.4.1.

In parallel experiments, fungal material from *A. fumigatus* was clearly stained with AB90-E8 (Figure 8A,B), whereas hyphae from *A. niger*, *A. nidulans* and *A. terreus* were not recognized (Figure 8C–E). Similar results were obtained from hyphal material grown in liquid culture and for all clinical isolates and wild type strains analysed. This demonstrates that AB90-E8 allows an expeditious identification of *A. fumigatus* directly from material taken from very small colonies.

#### 3.2.2. Direct Staining of Hyphae from Agar Plates—Tape Mount Immunostaining Method

An even more handy way to identify *A. fumigatus* directly from an agar plate is by using the tape mount immunostaining method that we previously introduced [39]. This method was established for analysis of fungal surface structures and can be easily replicated in every laboratory with common devices. We modified the original technique as described in Section 2.4.2. to facilitate its use in routine diagnostics.

The mAb AB90-E8 specifically stained hyphae of *A. fumigatus* and the closely related *A. fischeri*, but no other *Aspergillus* species, such as *A. nidulans*, *A. niger*, *A. terreus*, *A. oryzae* or *A. giganteus* (Figure 9). AB90-E8 clearly stained the hyphal wall of *A. fumigatus* and *A. fischeri* (Figure 9C,D). There was no unspecific binding to the adhesive tape.

#### 3.2.3. Staining of Histological Samples

We next analyzed whether AB90-E8 can be used in immunohistology. In vitro experiments showed stability of the AB90-E8 antigen after formaldehyde treatment, but we found that AB90-E8 is unable to stain hyphae in sections of paraffin-embedded samples. This was tested in paraffin sections from human aspergilloma of the lung and paranasal sinus from a patient with a PCR-proven *A. fumigatus* infection. In cryosections, however, AB90-E8 recognized *A. fumigatus* hyphae. This is shown in Figure 10 with biopsy material from a human aspergilloma of the maxillary sinus (*A. fumigatus* infection had been proven by PCR in the course of clinical diagnostics). The standard histological staining, Grocott’s silver staining (Figure 10C) and HE staining (Figure 10D), enabled the detection of filamentous fungi, and staining with the galactomannan-specific antibody L10-1 (Figure 10B) confirmed the identification of *Aspergillus* spp. hyphae. An additional staining with AB90-E8 allowed a specific labelling of *A. fumigatus* (Figure 10A).

## 4. Discussion

Diagnostic investigation of clinical cases suspected for IA is a challenging endeavor. Unspecific symptoms, such as fever of unknown origin, combined with an antibiotic treatment can indicate IA, especially in high-risk patients. The ESCMID-ECMM-ERS guidelines therefore recommend CT scans and bronchoscopy with bronchoalveolar lavage in those patients [41]. To ensure a correct diagnosis, direct microscopy, histopathology and culture are the current methods of choice. Brighteners such as Blankophor and Calcofluor White can improve the detection rate by visualizing the chitineous cell wall under UV light, but they are not specific to the fungal class [42,43,44]. Incubation at higher temperatures on selective agar can help to specifically isolate *A. fumigatus*, but this is not a rapid method [45]. The monoclonal antibody L10-1, which we used in this study as a control, as well as the commercially available monoclonal antibodies WF-AF-1 and EB-A1, label the cell wall polysaccharide galactomannan in immunostainings, but again are not specific for *A. fumigatus* [38,46,47,48].

The cell wall of *A. fumigatus* is a complex structure consisting of different carbohydrates and proteins. Antibodies that are reactive with hyphal surface antigens usually belong to the IgM subclass and recognize carbohydrate antigens [38,48,49]. Thornton (2008) described a monoclonal IgG_3_ antibody named JF5 that detects an unknown glycoprotein on the hyphal surface of *A. fumigatus* [50]. Interestingly, this antigen is released by the fungus and is detectable via a lateral flow device as well as an enzyme-linked immunosorbent assay (ELISA; Euroimmun Medizinische Labordiagnostika AG) for the diagnosis of IA [50,51,52]. The antigen of the AB90-E8 antibody is still unknown and the subject of ongoing research. The IgG subclass and the sensitivity to proteinase K digestion suggest that AB90-E8 recognizes a hyphal surface protein. The strong impact of the *glf*A deletion on the presence and distribution of the AB90-E8 antigen nevertheless suggests that certain glycostructures and in particular galactomannan seem to be important for a strong and persistent anchoring of the antigen in the cell wall. This notion is also in line with the partial loss of the AB90-E8 antigen from the surface of fludioxonil-treated cells, which are known to shed most of their surface-accessible galactomannan [40]. The particularly strong staining of the hyphal tips of Δ*glf*A mutant hyphae is a striking but explainable finding. The cellular apparatus that transports molecules to the hyphal surface is concentrated in the apical region [53]. The fact that AB90-E8 staining is focused at the hyphal tip suggests that the antigen is unable to persist on the surface for longer times, either because it is released or degraded.

Up to now, the Platelia ELISA (Platelia Aspergillus, Bio-Rad, USA) is the most commonly used method. Based on the galactomannan-antibody EB-A1, it is not specific for *A. fumigatus* and also detects other *Aspergillus* species as well as other filamentous fungi.

Currently, the only standard method that is widely used in diagnostic laboratories to specifically detect *A. fumigatus* is PCR. Another approach to distinguish *Aspergillus* isolates at the species level is to analyze the proteomics by using MALDI-TOF (matrix-assisted laser desorption ionization time-of-flight mass spectrometry) [54,55]. Even cryptic and rare species can, in principle, be detected by MALDI-TOF [56,57]. However, the generation of suitable samples from hyphae is troublesome, which, in combination with a lack of standardization due to inconsistent databases, still hinders a general and widespread use of MALDI-TOF in the clinical fungal diagnostics of filamentous fungi.

Microscopy and culture therefore remain the standard techniques for direct pathogen detection [58]. Morphological features (macroscopic: colony color and sporulation; microscopic: size and shape of conidia and hyphae, septation, pigmentation) and staining behavior in histology may lead to the identification of the fungal species.

The monoclonal antibody AB90-E8 specifically detects *A. fumigatus* and can be used for immunostainings of fungal culture material as soon as incipient colonies are available. This offers a great time advantage to the other commonly available diagnostic methods and classical approaches. Applied on frozen sections, it furthermore enables an intraoperative analysis of biopsy material and gives a lead in the diagnosis of IA.

As frozen sections are not usually prepared for routine histopathological examination in fungal diagnostics, the number of samples that were available in this study was limited to two. More tests are clearly required to determine the specificity and sensitivity in fresh frozen section analysis with AB90-E8.

Based on the results of this study, we can unreservedly recommend the use of AB90-E8 for a fast and specific identification of *A. fumigatus* by direct immunostaining of fungal culture material. The immunostaining techniques used in this study represent versatile and early tools to either confirm an infection with *A. fumigatus* or to prompt further research to investigate the potential presence of AmB-resistant non-fumigatus strains. Thus, the monoclonal antibody AB90-E8 gives an advantage in the often occuring race against time in high-risk patients with IA.

## Figures and Tables

**Figure 1 jof-09-00622-f001:**
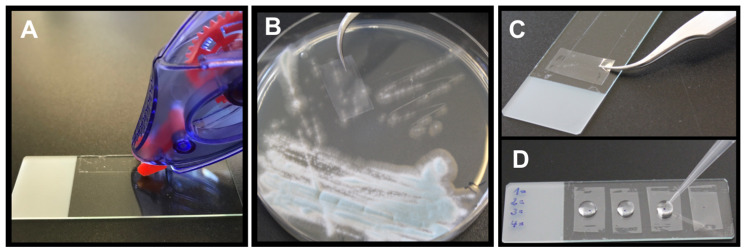
The modified tape mount immunostaining method. (**A**) a double-side sticky glue roller is applied to the upper and lower margin of an object slide; (**B**) the sticky side of transparent adhesive tape is pressed on early fungal colonies of an agar plate; (**C**) the non-sticky side of the adhesive tape is adhered to the glue stripes on the object slide; (**D**) multiple samples can be applied to one object slide. The centre of the region of interest is marked with a black dot and incubated consecutively with 50 µL of blocking solution, first and second antibody, and CFW with washing steps in between.

**Figure 2 jof-09-00622-f002:**
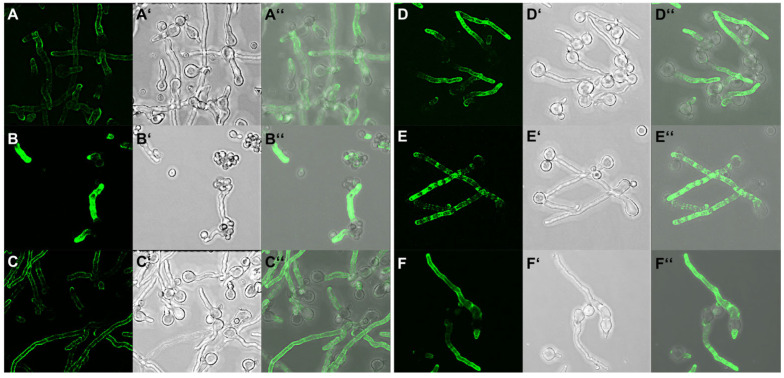
*A. fumigatus* strain AfS35 grown on glass cover slips in different media labelled with AB90-E8 (green). (**A**) AMM, (**B**) BSA medium, (**C**) HEPES-buffered RPMI-1640 medium, (**D**) Sabouraud, (**E**) SD and (**F**) YG medium. The images in (**A’**–**F’**) show the corresponding bright field images and (**A’’**–**F’’**) the respective overlays.

**Figure 3 jof-09-00622-f003:**
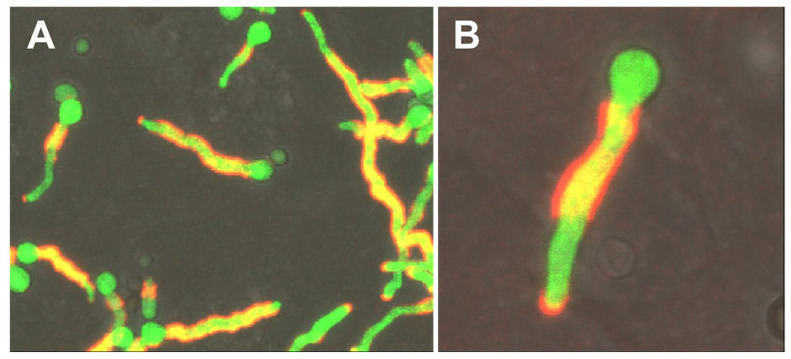
GFP-labelled *A. fumigatus* strain AfS35 (green) grown on A549 epithelial cells and stained with AB90-E8 (red). An overview is shown in (**A**), (**B**) shows one hypha at higher magnification.

**Figure 4 jof-09-00622-f004:**
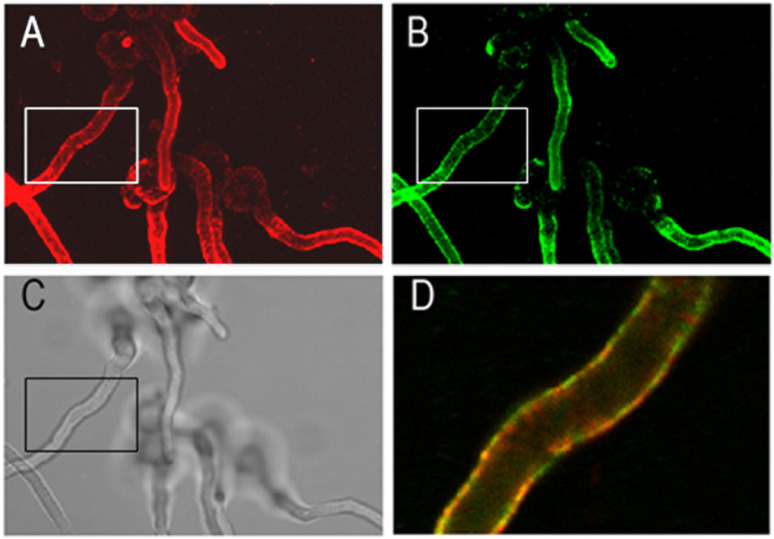
Co-staining with AB90-E8 and the galactomannan-specific antibody L10-1. Hyphae of *A. fumigatus* strain AfS35 were stained with L10-1 (**A**) and AB90-E8 (**B**). The corresponding bright field image is shown in (**C**). Panel (**D**) shows an overlay of both stainings. The corresponding region is indicated in (**A**–**C**). (**A**,**B**,**D**) are maximum projections of a stack of confocal images.

**Figure 5 jof-09-00622-f005:**
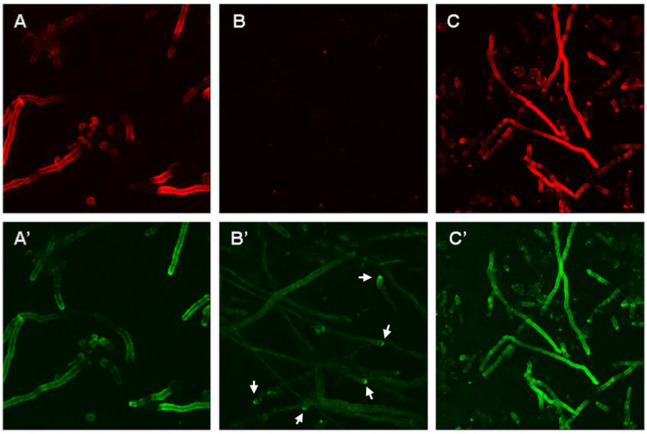
AB90-8 staining is influenced by galactomannan. We compared hyphae of the *Δglf*A mutant (**B’**), its parental strain D141 (**A’**) and the complemented mutant (**C’**) using immunofluorescence. (**A**–**C**) Staining with the galactomannan-specific antibody L10-1 demonstrates that the mutant lacks galactomannan (**B**) and this defect is restored in the complemented strain (**C**). Staining with AB90-E8 led to a more or less homogenous staining of wild type hyphae (**A’**), but a weaker staining of *Δglf*A mutant hyphae. Only the tips of the mutant hyphae showed a strong decoration with AB90-E8 (white arrows). Hyphae of the complemented mutant showed a pattern that resembled that of the wild type. All panels show maximum projections of several confocal images. The same exposure times were used for all pictures.

**Figure 6 jof-09-00622-f006:**
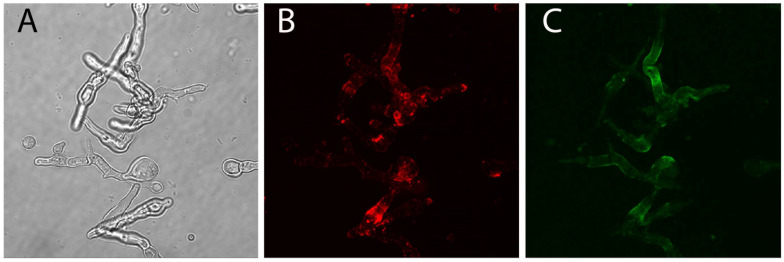
Germlings of *A. fumigatus* were treated with 2 μg fludioxonil per ml for 4 h and then stained with L10-1 (**B**) and AB90-8 (**C**). The corresponding bright field image is shown in (**A**).

**Figure 7 jof-09-00622-f007:**
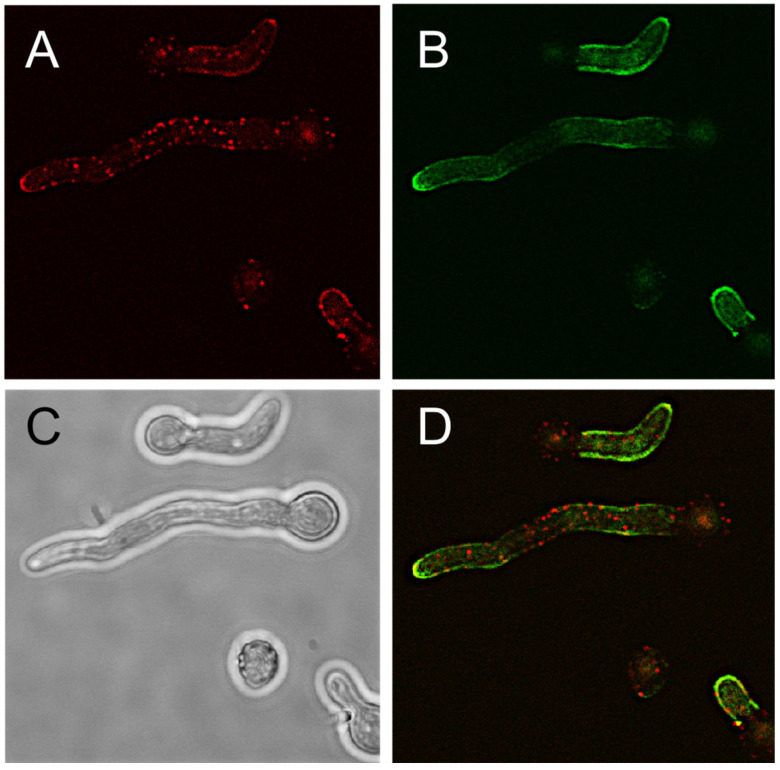
Co-Staining of *A. fumigatus* hyphae with AB90-E8 (green, (**B**)) and anti-MP1 (red, (**A**)). The overlay and the corresponding bright field image are shown on (**D**) and (**C**), respectively.

**Figure 8 jof-09-00622-f008:**
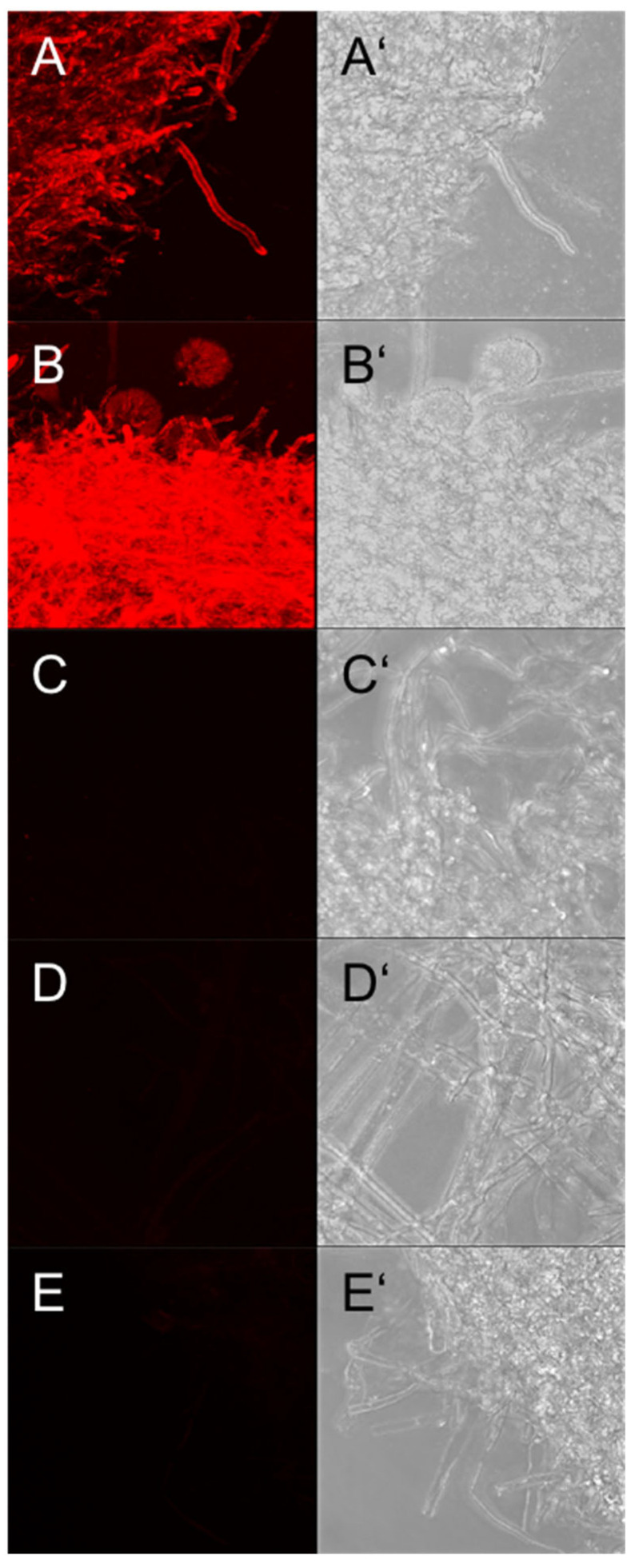
Fungal material directly picked from an agar plate with tweezers: (**A**) *A. fumigatus* strain D141, (**B**) *A. fumigatus* strain ATCC46645, (**C**) *A. terreus* strain T9, (**D**) *A. nidulans* strain A26, (**E**) *A. niger* strain DSM737. Immunostaining with AB90-E8 is shown in (**A**–**E**) and the corresponding bright field images are depicted in (**A’**–**E’**). The same exposure times were used for (**A**–**E**).

**Figure 9 jof-09-00622-f009:**
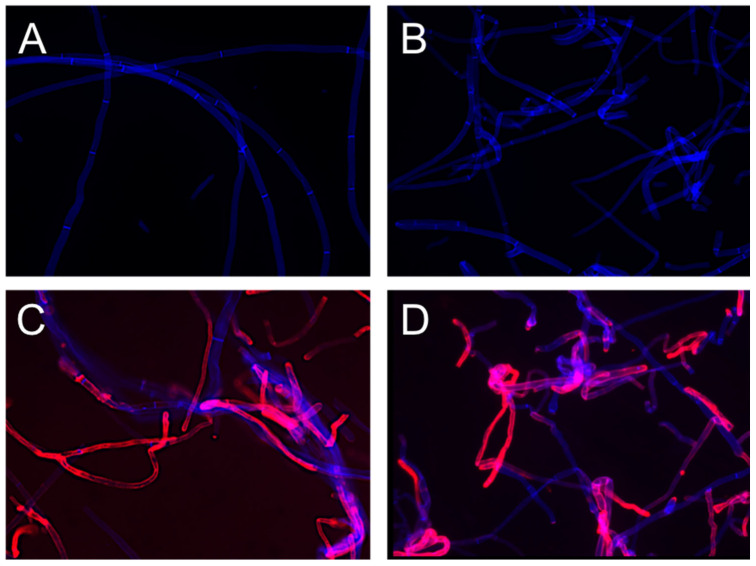
AB90-E8 immunostainings of fungal material stuck to adhesive tape: (**A**) *A. niger*, (**B**) *A. nidulans*, (**C**) *A. fischeri*, (**D**) *A. fumigatus*. CFW stained the fungal cell wall (blue), whereas AB90-E8 labelled only *A. fischeri* and *A. fumigatus* (red).

**Figure 10 jof-09-00622-f010:**
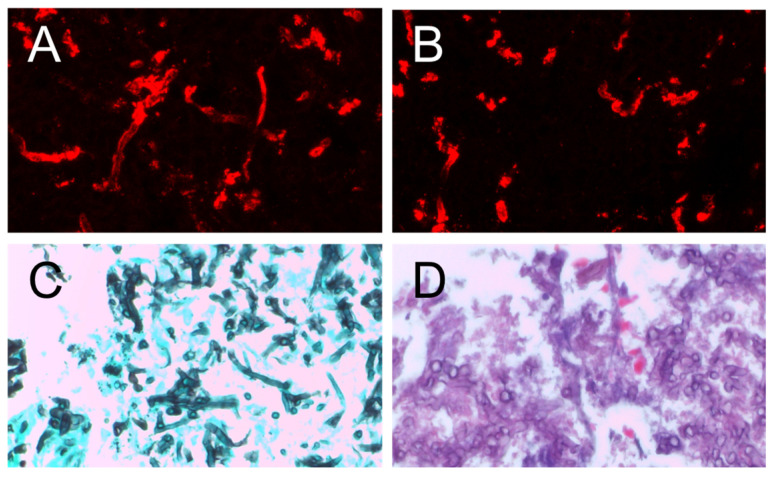
Fresh frozen section of an aspergilloPleasma of the maxillary sinus: (**A**) immunostaining with AB90-E8, (**B**) immunostaining with L10-1, (**C**) Grocott’s silver staining, (**D**) HE staining.

## Data Availability

Not applicable.

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
