# Peer review of "The Novel Monoclonal IgG1-Antibody AB90-E8 as a Diagnostic Tool to Rapidly Distinguish Aspergillus fumigatus from Other Human Pathogenic Aspergillus Species"

_jof, 2023, doi:10.3390/jof9060622_

Round 1
Reviewer 1 Report
In the manuscript entitled "The Novel Monoclonal IgG1-Antibody AB90-E8 as a Diagnostic Tool to Rapidly Distinguish Aspergillus fumigatus from other Human Pathogenic Aspergillus Species." Kakoschke et al. describe the use of a previously generated monoclonal antibody for specific immunofluorescence staining of A. fumigatus to distinguish it from other molds causing invasive aspergillosis. It seems that the antibody has already been published recently. Nevertheless, in the present manuscript the authors provide further characterization of the antibody and nice technical advice regarding sample preparation for immunostainings.
The manuscript is well written with an informative introduction and the presented results support the conclusions drawn. Therefore, I only have some minor issues:
L 102ff: Please include reference/detailed recipe or ordering details for fungal media.
L 340: Please remove one period.
L 415/417/420: Please use MALDI-TOF.
L 134: Tween 20?
L 141: Please include ordering details for fludioxonil.
L 179f: Tape strip?
Figure 5B; Figure 6C/D/E: I assume that exposure times were similar or identical to the panels showing strong signals. If so, this information should be included in the figure legends. Alternatively, pictures derived from overexposure might be displayed.
Author Response
First of all, we like to thank the reviewers for their time and valuable remarks that helped us to improve the manuscript. We have revised the text according to the suggestions and provide a detailed point-by-point response below. All changes to the manuscript are marked up using the “Track Changes” function of MS Word.
We hope that the revised manuscript is now acceptable as a publication in the Journal of Fungi.
Sincerely,
Tamara Kakoschke
Comments from Reviewer 1:
In the manuscript entitled "The Novel Monoclonal IgG1-Antibody AB90-E8 as a Diagnostic Tool to Rapidly Distinguish Aspergillus fumigatus from other Human Pathogenic Aspergillus Species." Kakoschke et al. describe the use of a previously generated monoclonal antibody for specific immunofluorescence staining of A. fumigatus to distinguish it from other molds causing invasive aspergillosis. It seems that the antibody has already been published recently. Nevertheless, in the present manuscript the authors provide further characterization of the antibody and nice technical advice regarding sample preparation for immunostainings.
The manuscript is well written with an informative introduction and the presented results support the conclusions drawn. Therefore, I only have some minor issues:
Comment 1: L 102ff: Please include reference/detailed recipe or ordering details for fungal media.
Answer 1: We have included this information in the revised text.
Comment 2: L 340: Please remove one period.
Answer 2: Done as suggested.
Comment 3: L 415/417/420: Please use MALDI-TOF.
Answer 3: Thank you for this note. We changed it.
Comment 4: L 134: Tween 20?
Answer 4: Yes. Added.
Comment 5: L 141: Please include ordering details for fludioxonil.
Answer 5: Done as suggested.
Comment 6: L 179f: Tape strip?
Answer 6: We added “strip”.
Comment 7: Figure 5B; Figure 6C/D/E: I assume that exposure times were similar or identical to the panels showing strong signals. If so, this information should be included in the figure legends. Alternatively, pictures derived from overexposure might be displayed.
Answer 7: We appreciate your attention. We have used the same exposure times for all panels and included this information in the legends of Figure 5, Figure 8
Comments from Reviewer 2:
In search for a novel rapid and specific diagnostic technique to identify A. fumigatus in clinical samples, Kakoschke et al. investigated the potential of the monoclonal antibody AB90-E8 for immunostaining using fungal culture material or histological samples.
Although the specific identification of intrinsically antimicrobial-resistant Aspergillus species would be more valuable, the idea presented in this study is of a good scientific interest.
The manuscript is well written and very easy to follow.
However, a few minor points should be addressed.
Comment 1: Line 102-106: The indications for the use of alternative culture media are not clearly specified in the following paragraphs (only the use of AMM is described in 2.3.2.).
Comment 2: Line 131: Specify the medium and the incubation time.
Answer 1 and 2: We have now specified the media that were used in the different experiments. This is also mentioned in the sentence the comment 2 is referring to.
Comment 3: Line 142: Remove the double dot after 2.3.1 or add a dot after 2.3.1 in line 148.
Answer 3: We have added a dot in line 148.
Comment 4: Line 252: Specify if images of the same A. fumigatus strains are presented in Fig.2.
Answer 4: Thank you for your note. Images of the same A. fumigatus strain AfS35 can be seen in Figure 2. We have added this detail.
Comment 5: Line 261-265: it is not clear if the results on the immunostaining reactivity were obtained in the presented study (as it seems as the same Aspergillus species are described in M&M) or in a previous study (as it seems from the bibliographic references reported in line 264 and 265).
Answer 5: The information given in line 261 – 265 was already published by us in [24]. To clarify this, we have revised the text slightly.
Comment 6: A comment on the results of paragraph 3.1 “Characterization of the monoclonal antibody AB90-E8” is missing in the Discussion section.
Are there any hypotheses on the prevalent localization of AB90-E8 on the hyphal tips, particularly in absence of galactofuranose? Treatment with proteinase K destroys the AB90-E8 antigen, however the pattern of AB90-E8 staining in the complemented ΔglfaA mutant seems to suggest an influence of galactofuranose on antibody binding. May this be linked to antigen modifications during hyphal maturation?
Answer 6: To address this point we have introduced the following sentences in the Discussion part: The antigen of the AB90-E8 antibody is still unknown and the subject of ongoing research. The IgG subclass and the sensitivity to proteinase K digestion suggest that AB90-E8 recognizes a hyphal surface protein. The strong impact of the glfA deletion on the presence and distribution of the AB90-E8 antigen nevertheless suggests that certain glycostructures and in particular galactomannan seem to be important for a strong and persistent anchoring of the antigen in the cell wall. This notion is also in line with the partial loss of the AB90-E8 antigen from the surface of fludioxonil-treated cells, which are known to shed most of their surface-accessible galactomannan [40]. The particular strong staining of the hyphal tips of ΔglfA mutant hyphae is a striking, but explainable finding. The cellular apparatus that transports molecules to the hyphal surface is concentrated in the apical region [53]. The fact that the AB90-E8 staining is focused at the hyphal tip suggests that the antigen is unable to persist on the surface for longer times, either because it is released or degraded.
Comment 7: Line 424-424: Please reformulate the sentence as in its present form seems that the author state that observation of fungal microscopic morphological features “can at least SUGGEST the fungal species”.
This is not exactly true as the observation of microscopic structures such as spores and spore-bearing cells may lead to mold species identification.
The main problem with this classical approach is that this process takes longer than immunostaining of incipient colonies.
Answer 7: We agree and have changed the wording accordingly.
Comment 8: Lines 437-439: I do not agree that the presented immunostaining techniques may represent a tool “to decide when to drop AmB treatment”.
In fact, this technique allows the differentiation between A. fumigatus and non-A. fumigatus infections, but it doesn’t identify possible AmB-resistant strains among non-A. fumigatus.
Nevertheless, I agree that this may represent a rapid tool to either confirm an A. fumigatusinfection or to prompt further research to investigate the presence of AmB resistant strains.
Answer 8: We agree and have change the sentences accordingly.

Reviewer 2 Report
In search for a novel rapid and specific diagnostic technique to identify A. fumigatus in clinical samples, Kakoschke et al. investigated the potential of the monoclonal antibody AB90-E8 for immunostaining using fungal culture material or histological samples.
Although the specific identification of intrinsically antimicrobial-resistant Aspergillus species would be more valuable, the idea presented in this study is of a good scientific interest.
The manuscript is well written and very easy to follow.
However, a few minor points should be addressed.
Line 102-106: The indications for the use of alternative culture media are not clearly specified in the following paragraphs (only the use of AMM is described in 2.3.2.).
Line 131: Specify the medium and the incubation time.
Line 142: Remove the double dot after 2.3.1 or add a dot after 2.3.1 in line 148.
Line 252: Specify if images of the same A. fumigatus strains are presented in Fig.2.
Line 261-265: it is not clear if the results on the immunostaining reactivity were obtained in the presented study (as it seems as the same Aspergillus species are described in M&M) or in a previous study (as it seems from the bibliographic references reported in line 264 and 265).
A comment on the results of paragraph 3.1 “Characterization of the monoclonal antibody AB90-E8” is missing in the Discussion section.
Are there any hypotheses on the prevalent localization of AB90-E8 on the hyphal tips, particularly in absence of galactofuranose? Treatment with proteinase K destroys the AB90-E8 antigen, however the pattern of AB90-E8 staining in the complemented ΔglfaA mutant seems to suggest an influence of galactofuranose on antibody binding. May this be linked to antigen modifications during hyphal maturation?
Line 424-424: Please reformulate the sentence as in its present form seems that the author state that observation of fungal microscopic morphological features “can at least SUGGEST the fungal species”.
This is not exactly true as the observation of microscopic structures such as spores and spore-bearing cells may lead to mold species identification.
The main problem with this classical approach is that this process takes longer than immunostaining of incipient colonies.
Lines 437-439: I do not agree that the presented immunostaining techniques may represent a tool “to decide when to drop AmB treatment”.
In fact, this technique allows the differentiation between A. fumigatus and non-A. fumigatus infections, but it doesn’t identify possible AmB-resistant strains among non-A. fumigatus.
Nevertheless, I agree that this may represent a rapid tool to either confirm an A. fumigatus infection or to prompt further research to investigate the presence of AmB resistant strains.
Author Response

(The authors gave the same response as above.)
